# Harnessing Machine Learning for Classifying Economic Damage Trends in Transportation Infrastructure Projects

**Junseo Bae** [1] **, Sang-Guk Yum** [2] **and Ji-Myong Kim** [3,*]

1  School of Computing, Engineering and Physical Sciences, University of the West of Scotland,
   Paisley PA1 2BE, UK; junseo.bae@uws.ac.uk
2  Department of Civil Engineering, Gangneung-Wonju National University, Gangneung 25457, Korea;
   skyeom0401@gwnu.ac.kr
3  Department of Architectural Engineering, Mokpo National University, Mokpo 58554, Korea
*  Correspondence: jimy6180@gmail.com

**Abstract:** Given the highly visible nature, transportation infrastructure construction projects are often exposed to numerous unexpected events, compared to other types of construction projects. Despite the importance of predicting financial losses caused by risk, it is still difficult to determine which risk factors are generally critical and when these risks tend to occur, without benchmarkable references. Most of existing methods are prediction-focused, project type-specific, while ignoring the timing aspect of risk. This study filled these knowledge gaps by developing a neural network-driven machine-learning classification model that can categorize causes of financial losses depending on insurance claim payout proportions and risk occurrence timing, drawing on 625 transportation infrastructure construction projects including bridges, roads, and tunnels. The developed network model showed acceptable classification accuracy of 74.1%, 69.4%, and 71.8% in training, cross-validation, and test sets, respectively. This study is the first of its kind by providing benchmarkable classification references of economic damage trends in transportation infrastructure projects. The proposed holistic approach will help construction practitioners consider the uncertainty of project management and the potential impact of natural hazards proactively, with the risk occurrence timing trends. This study will also assist insurance companies with developing sustainable financial management plans for transportation infrastructure projects.

**Keywords:** transportation infrastructure; economic damage; financial loss; insurance claim payout; risk occurrence timing; machine learning; neural network; data classification; risk assessment

## 1. Introduction

### 1.1. Point of Departure

The third-party liability insurance has been commonly adopted by the owners of large-scale construction projects, in order to cover economic damages caused by construction operation, project management, or other external risks [1]. In general, risk is interpreted as the probability of losses that would be caused by undesirable events, such as damage, vulnerability, or disaster in a certain region during particular time periods [2–5]. To reduce economic damages, it is pivotal to use proper risk assessment methods and thus to contribute to improving sustainability in project management. Given the significance of sustainable risk management, many research efforts were made by estimating economic damages in broad ranges of construction projects [1,6–16].

Among various construction projects, transportation infrastructure projects need to paid more attention because these are larger, more complex, and mostly capital-intensive, compared to others. Transportation infrastructure is well known as a key built asset to facilitate sustainable economic growth [17–20]. More specifically, as essential social and economic assets, transportation infrastructure systems (e.g., roads, bridges, tunnels) construct space, enhance the productivity of a nation by increasing the mobility and

responding to social needs. However, given the highly visible nature, transportation infrastructure construction projects are more often exposed to numerous unexpected events (i.e., risk), compared to other construction projects.

For better risk management in transportation infrastructure projects, many studies have focused on predicting the value of economic damages in certain types of projects. Some researchers identified risks in bridge construction projects, including safety, traffic conditions, shortage of labor and materials, adverse weather conditions, financial risk, equipment risk, health and safety issues, construction and management risk, contractual issues [21–25]. In addition, bridge projects were examined statistically to estimate economic damages depending on project information (i.e., types of superstructures and foundations, the scope of project, construction cost and schedule, construction methods, company's reputation) and environmental risk factors based on flood and typhoon records [6,9]. More recently, the impact of natural hazard-driven risk on the financial loss was assessed based on tunnel projects. To this end, wind speeds, flooding occurrences, and rain falls were incorporated into a multiple regression analysis [1]. Despite many research efforts, most of previous studies are project type-specific. In this sense, it is still not easy to determine which risk factors are generally critical and when these risks tend to occur, without proven references to be benchmarked. Hence, it should be also highlighted that post-assessment of economic damages is crucial to provide benchmarkable references and thus develop more effective and sustainable risk management strategies for transportation construction project plans in the future.

To achieve benchmarkable references, classifying generalizable patterns in economic damages into certain risk indicators. Responding to this need, it is acknowledged that hidden patterns within a set of data can be uncovered through a data mining process [26]. Compared to prediction approaches, a classification method is intended to build a model that can classify data points by evaluating accuracy of characteristics of data [27]. However, a solid literature review concludes that most of previous studies focused on predicting financial losses associated with various risk factors that cannot universal for different types of transportation infrastructure projects.

In the pursuit of classifying the trends, it would be more beneficial to know about risk occurrence timing trends in conjunction with risk factors and financial losses occurred in each different type of transportation infrastructure projects because economic damages can be evaluated deeply and systematically at the temporal scale. However, through a thorough literature review, it was found that very little known about timing aspects. Overall, knowledge about a holistic approach to classifying generalizable trends in financial losses by key indicators of economic damage is still largely missing.

### 1.2. Research Objective and Method

To fill the gaps in existing knowledge delineated above, this study attempted to scientifically classify economic damages in major transportation infrastructure projects. More specifically, the main objective of this study is to develop a neural network-driven machine-learning classification model that can categorize causes of financial losses depending on insurance claim payouts caused by risks and their occurrence timing in three different major types of transportation infrastructure projects, such as bridges, roads, and tunnels. Machine learning techniques have become the most widely-used method for either prediction or classification by finding hidden patterns of data, specifically aiming at unlocking the complexity and nonlinearity of data [28]. Especially, in the context of assessing financial loss or economic development, it has been underlined that the use of machine learning techniques is effective to improve predictability [29,30]. Compared to prediction approaches, a machine-learning classification method aims to automatically maximize the classification accuracy by certain indicators or groups in order to assess the performance of any kind of systems [26]. Given the strength, machine-learning classification approach was adopted for this study. More specifically, the objective of this study was achieved by the following five steps:

1.  Collect construction insurance and project data from 625 transportation infrastructure projects completed over the past 15 years in South Korea, which consist of bridge, road, tunnel construction projects;
2.  Identify causes of financial losses and cluster the causes to set the learning database;
3.  Explore various neural network alternatives and shortlist the top five;
4.  Assess the classification accuracy of the shortlisted network alternatives and select the most feasible classification model;
5.  Classify the generalized financial loss clusters by insurance claim payout proportions and risk occurrence timing per project type.

The following is the assumption of this research:

It was assumed that a certain point when claim payout was made is same as the timing when risk occurred. In other words, a few days of gaps between risk occurrence and paid-out amounts of construction insurance were assumed to insignificantly affect the classification performance.

## 2. Materials and Methods

### 2.1. Economic Damage Indicators for Measurement: Magnitude and Timing Aspects

For this study, the following indicators were used to learn and classify generalized economic damages in three different types of transportation infrastructure projects:

-   Claim payout ratio (CPR);
-   Occurrence timing (OT).

CPR was used to examine the proportion of insurance claim payout amounts, against the total amount of contract insurance allocated. It is defined as the following Equation (1):

$$CPR = \frac{Claim\ Payout\ Amount\ (KRW)}{Total\ Amount\ of\ Insurance\ Allocated\ (KRW)} \tag{1}$$

As a way to interpret CPRs, if a CPR is less than one, a claim caused by project risks (e.g., management, natural hazards, etc.) was covered sufficiently from the allocated amount of insurance. On the other hand, the value of CPR exceeding one means that project risks were severe, resulting in over-paid insurance amounts.

In addition, OT was used to capture the trends in risk occurrences at the time scale, which provides generalizable knowledge when certain causes of financial losses occurred. To this end, OT is defined as seen in Equation (2):

$$OT = \frac{(Claim\ Payout\ Date - Contract\ Start\ Date)}{(The\ Original\ Contract\ End\ Date - Contract\ Start\ Date)} \tag{2}$$

For example, the OT value of 0.3 indicates that the financial loss caused by a certain project risk occurred since 30% of the project schedule has processed. If the value of OT is greater than 1, it represents that the claim was accepted for risk occurred, based on the contract adjustment. This case would be acknowledged that the planned project schedule was delayed for some unforeseen reasons (e.g., change orders, idle time based on the proactive plan adjustment, due to forecasted severe weather conditions).

### 2.2. Descriptive Analysis

The insurance- and project-related data sets were collected from 625 transportation infrastructure projects completed in South Korea, which consist of bridge, road, tunnel construction projects. The collected data sets include the total amount of construction insurance, insurance claim payout amounts, contract start and end dates, project types, risk occurrence dates, types of causes of financial losses. In transportation infrastructure projects adopted in this study, 14 different causes of financial losses were identified:

-   Careless;
-   Collapse;

- Damage;
- Fire & explosion;
- Slips, trips & falls;
- Soil settlement;
- Stolen;
- Flooding;
- Heavy rain;
- Heavy snow;
- Hurricane;
- Storm;
- Thunderstroke;
- Others.

The identified risks were then classified into three different clusters for the purpose of improving the classification performance of the proposed machine learning model: (1) management risk; (2) natural hazards; (3) miscellaneous (misc.) risk. Figure 1 summarizes the proportions of risks identified in the whole set of data, which is classified into three different clusters by project types (i.e., bridge, road, tunnel). As a snapshot of causes of financial losses, in the management risk group, it was found that bridge and tunnel projects were often affected by damage and slips, trips, and falls. In this study, damage indicates financial losses particularly caused by equipment malfunction, mechanical or construction defect. Damage (15.36%) and soil settlement (11.52%) were identified as the most frequent causes of financial loss in road projects. When it comes to natural hazards, all the types of transportation projects were often affected by heavy rainfalls. However, it was found that historical natural hazard records in bridge and tunnel projects hold a relatively smaller portion, compared to road projects.

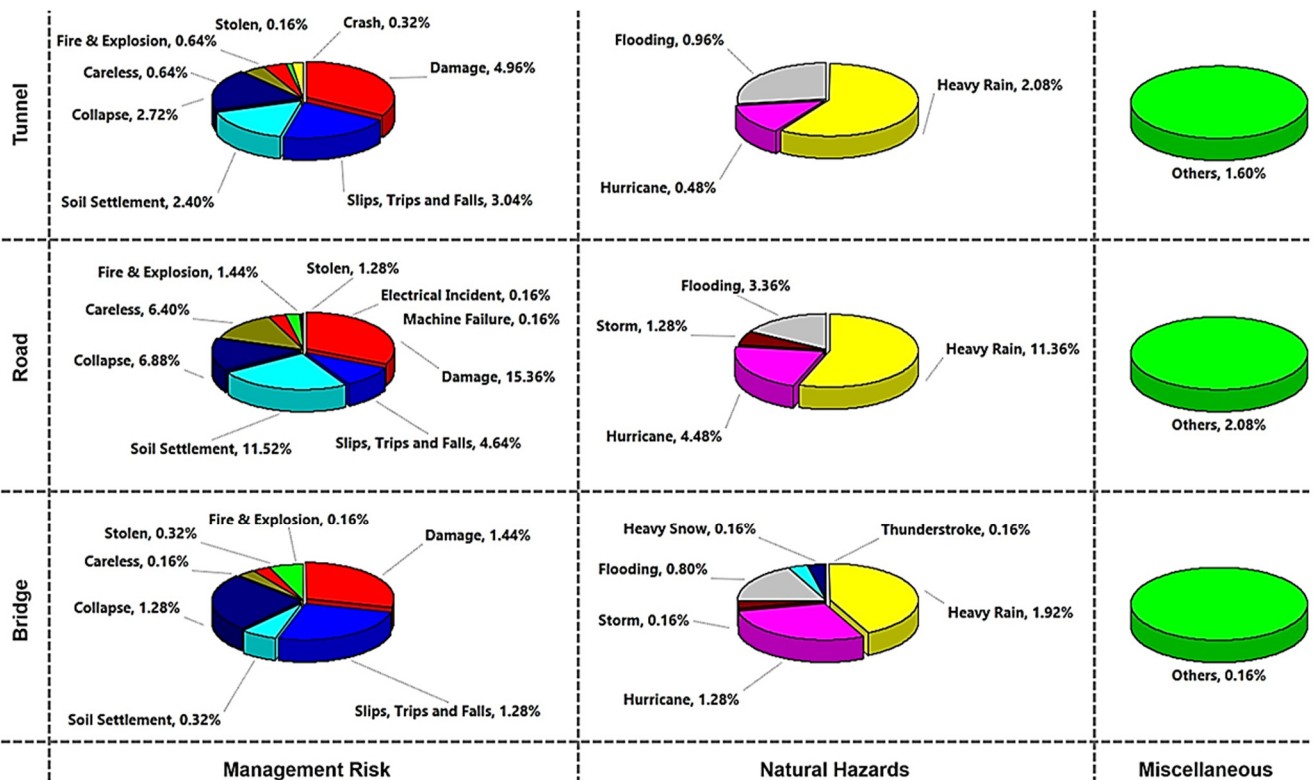

**Figure 1.** Financial loss clusters: Causes of financial loss (i.e., risk) by project types.

Based on the financial cause clusters, Table 1 summarizes variables used in the machine learning classification process. In detail, to classify economic damages, claim payout ratios and risk occurrence timing drawing on 625 data points were used as continuous input variables. Meanwhile, three different types of transportation infrastructure projects were employed.

Then, to improve the accuracy performance of learning model alternatives, temporal indicators at the seasonal scale were mapped with the corresponding date of risk occurrence date (i.e., claim payout date). The addition of temporal indicators was supported by the sample size of the financial cause clusters by project types during spring, summer, fall, and winter seasons, shown in Figure 2. Initially, it was expected that the only natural hazards cluster is significantly affected by summer season, considering the effect of heavy rainfalls shown in Figure 1. However, for all the three clusters, most of risks were identified commonly during four seasons. Hence, temporal indicators were assigned to the categorical input variable.

**Table 1.** Description of variables.

| Classification Model Variable (Data Type) | | Mean | Median | Minimum | Maximum | Std. Dev. |
|---|---|---|---|---|---|---|
| Inputs | Claim Payout Ratio (continuous) | 0.48716 | 0.17815 | 0.00062 | 9.60976 | 0.96602 |
| | Risk Occurrence Timing (continuous) | 0.60374 | 0.60000 | 0.00438 | 1.411713 | 0.29508 |
| | Construction Project Type (categorical) | • Bridge<br>• Road<br>• Tunnel | | | | |
| | Temporal Indicator (categorical) | • Spring (March-May)<br>• Summer (June-August)<br>• Fall (September-November)<br>• Winter (December-February) | | | | |
| Outputs | Financial Loss Cluster (categorical) | • Management Risk = [{careless}, {collapse}, {damage}, {fire & explosion}, {slips, trips & falls}, {soil settlement}, {stolen}]<br>• Natural Hazards = [{flooding}, {heavy rain}, {heavy snow}, {hurricane}, {storm}, {thunderstroke}]<br>• Miscellaneous = [others] | | | | |

### 2.3. Developing a Neural Network-Driven Machine-Learning Classification Model

As machine learning classification techniques, support vector machine (SVM) and neural networks have been most commonly used in various study areas [26,31]. It is well known that SVM is powerful to improve the generalization ability against nonlinear data, but it is difficult to interpret the results of learning outputs with no tangible shape of the trained model. In addition, it is underpinned by the similarity function (i.e., kernel), but there is no common solution to determine the optimal kernel that maximizes the distance between the nearest values in different groups. In contrast, neural networks are capable of providing a tangible model structure unveiled from hidden patterns, while effectively controlling highly nonlinear characteristics of data [26,31].

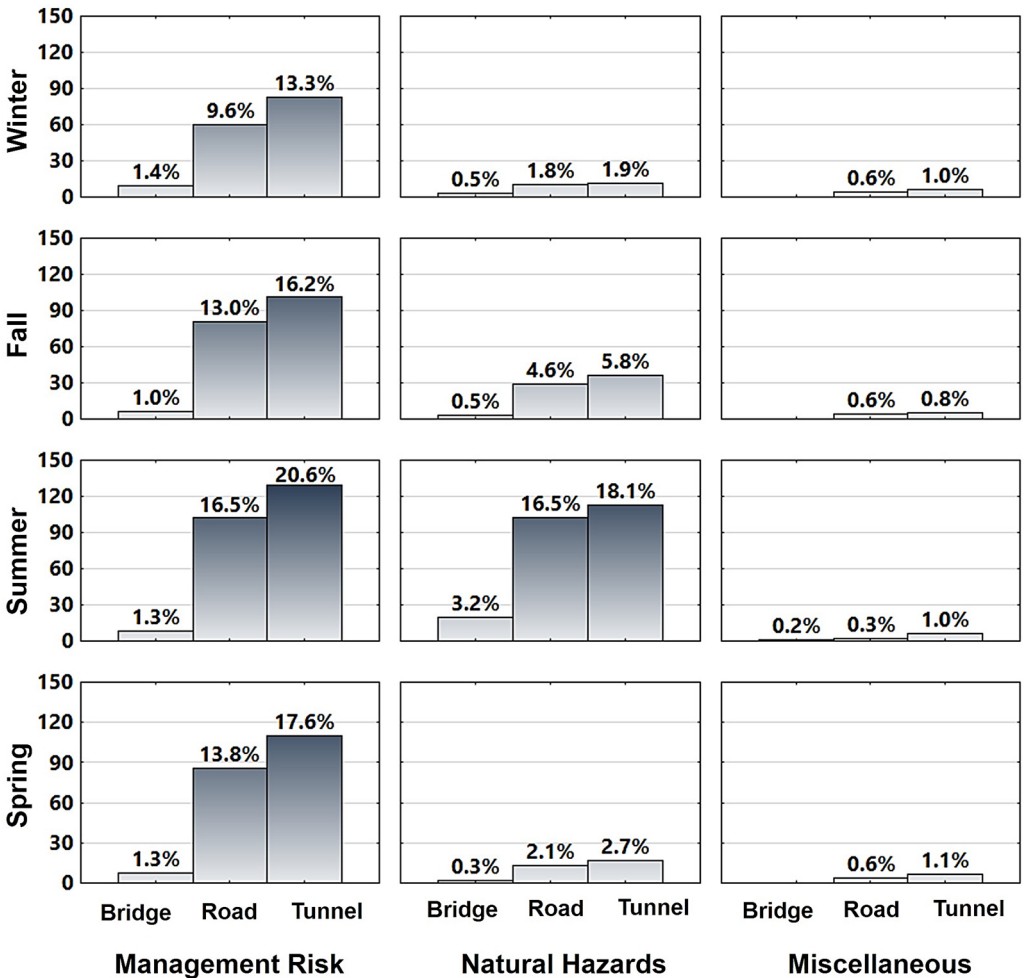

**Figure 2.** Cases of financial loss clusters by project types at the seasonal level.

Given the nonlinearity of data and different types of variables used in this study, a neural network technique was adopted to develop a classification model that can accurately categorize the financial cause clusters associated with claim payout proportions and risk occurrence timing drawing on major transportation infrastructure projects, such as bridges, roads, and tunnels. The network structure was firstly confirmed based on 9 input nodes and 3 output nodes. As shown in Table 1, 9 input variables include claim payout ratios, risk occurrence timings, project types including bridges, roads, tunnels, and four seasons, while 3 outputs indicate three different financial loss clusters based on management risk, natural hazards, and miscellaneous risks. The topology of the proposed classification model includes an input layer, one hidden layer, and an output layer.

### 2.3.1. Setting the Subsets of Data and Training Algorithm

It is acknowledged that the goodness of fit is affected by the complexity of the learning model on aspects of the number of hidden nodes and types of variables. To improve the learning performance, in general, the 80:20 split is widely used for training set (60% of training and 20% of cross-validation) and test set (20%). A training set plays a key role in monitoring the optimal connection weight that can minimize errors in the neural network. As a subset of training set, a cross-validation set is used to determine the optimal number of hidden nodes by detecting errors during training and then stopping the training properly. Then, the generalization ability of the fully trained network is assessed by the test set [32,33]. This split method is suitable when a large amount of data to train is available [34].

For this study, due to a smaller set of data used, a 70:30 holdout method was alternately adopted, which divides the original data into 70% training set and 30% test set. Within the training set, 30% of data was assigned randomly to the cross-validation set to determine the optimal number of hidden nodes. This alternate approach was intended to improve the learning performance and thus achieve more accurate classification outputs.

As stated earlier, the most feasible training process is determined by connection weights. The connection weights are unknown and thus estimated by a training algorithm [35]. The back propagation (BP) algorithm is well known as a training algorithm for neural network applications [35–37]. It trains networks by producing random weights and changing them iteratively, until the most feasible network is found. The standard BP algorithm often confined to the local minima, while converging slowly. In contrast, conjugate gradient (CG) and Broyden-Fletcher-Goldfarb-Shanno (BFGS) quasi-Newton methods are known as advanced BP algorithms, which avoid a local minima issue and converge faster than the standard BP algorithm [38]. It was underscored by some researchers that BFGS often converges faster than the CG [36–39]. Hence, this study adopted the BFGS algorithm to train the neural networks and classify economic damages in transportation infrastructure projects, on aspects of claim payout proportions by risk occurrence timing. The training process was conducted using the automated search engine in Statistica software version 13.3.

### 2.3.2. Training the Network Alternatives

As stated earlier, achieving the optimal number of hidden nodes is driven by a cross-validation set. The number of hidden nodes is dependent mainly on the complexity of network structure and types of variables to be trained. To search the most feasible number of hidden nodes for this study, the node numbers ranging from 1 to 200 were explored in the search module. Meanwhile, a total of 100 different network structure alternatives were considered associated with 25 different combinations of activation functions for hidden and output layers. Activation functions serve to transform input values and then determine the output of nodes processed. In detail, for this study, five different functions were applied for the training process (five different possible cases per layer): the identity function ($f(x) = x$); the exponential function ($f(x) = x$, *if* $x \geq 0$; $f(x) = \alpha[\exp(x) - 1]$, *if* $x < 0$); the hyperbolic tangent function ($f(x) = [\exp(x) - \exp(-x)]/[\exp(x) + \exp(-x)]$); the logistic function ($f(x) = [1 + \exp(-x)]^{-1}$); the sine function ($f(x) = \sin(x)$). To find a reasonable stopping point of the training process in each different network alternative, the maximum number of epochs (i.e., training cycles) was set to 500. By training all the network alternatives, five alternatives were retained as the shortlist.

### 3. Results

The results shown in Table 2 reveal that most of shortlisted network alternatives are reasonably accurate, holding about 70% accuracy of the classification. Among these, NN 9-70-3 BFGS 39 with the exponential function for the hidden activation and the hyper tangent (tanh) function for the output activation shows the highest accuracy in all the subsets of data, 74.104%, 69.38%, and 71.845% in training, cross-validation, and test sets, respectively. BFGS 39 means that the fully trained network was achieve at the 39th training cycle out of 500. With the highest classification accuracy, NN 9-70-3 was selected as the final model to classify financial loss clusters by claim payout ratios and risk occurrence timing in bridge, road, and tunnel construction projects. Accordingly, Figure 3 illustrates the structure of the final model.

**Table 2.** Shortlisted network alternatives.

| Neural Network Alternative | Accuracy (%) | | | Training Algorithm | Activation Function | |
| --- | --- | --- | --- | --- | --- | --- |
| | Training Set | Cross-Validation Set | Test Set | | Hidden Layer | Output Layer |
| 9-70-3 * | 74.104 | 69.380 | 71.845 | BFGS 39 | Exponential | Tanh |
| 9-154-3 | 72.908 | 65.241 | 70.053 | BFGS 21 | Tanh | Identity |
| 9-116-3 | 72.510 | 65.241 | 70.588 | BFGS 13 | Sine | Exponential |
| 9-79-3 | 72.908 | 65.241 | 70.054 | BFGS 22 | Sine | Tanh |
| 9-156-3 | 66.534 | 67.380 | 71.123 | BFGS 3 | Exponential | Sine |

\* The network structure selected as the final model.

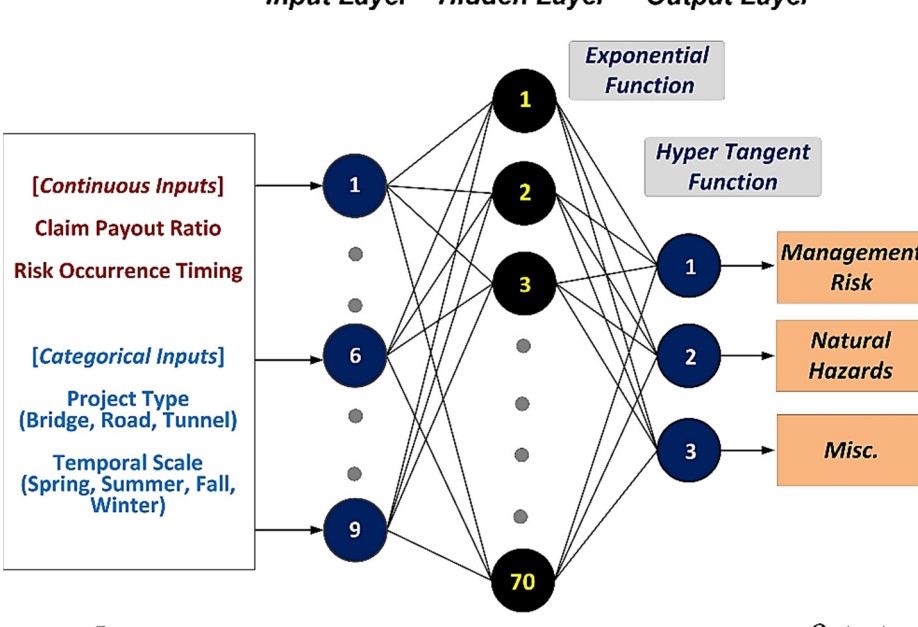

**Figure 3.** Final classification model: NN 9-70-3.

## 4. Discussion

The economic damage trends in transportation infrastructure projects that are learned from the developed neural network model were classified into the claim payout proportions and risk occurrence timing, using box-and-whisker plots. As depicted in Figure 4, the plots consist of the median values, the upper (75%) and lower (25%) quartiles, non-outlier ranges, and the mean values of generalized claim payout ratios under two different financial loss clusters (i.e., management risk, natural hazards) associated with risk occurrence timing. It should be noted that a small portion of the miscellaneous risk occurrence cases were not captured by the developed classification model. In addition, due to a smaller set of natural hazards occurred in tunnel projects, the corresponding outcomes were excluded from the generalized classification, which is aligned well with historical records of tunnel projects, seen in Figure 1.

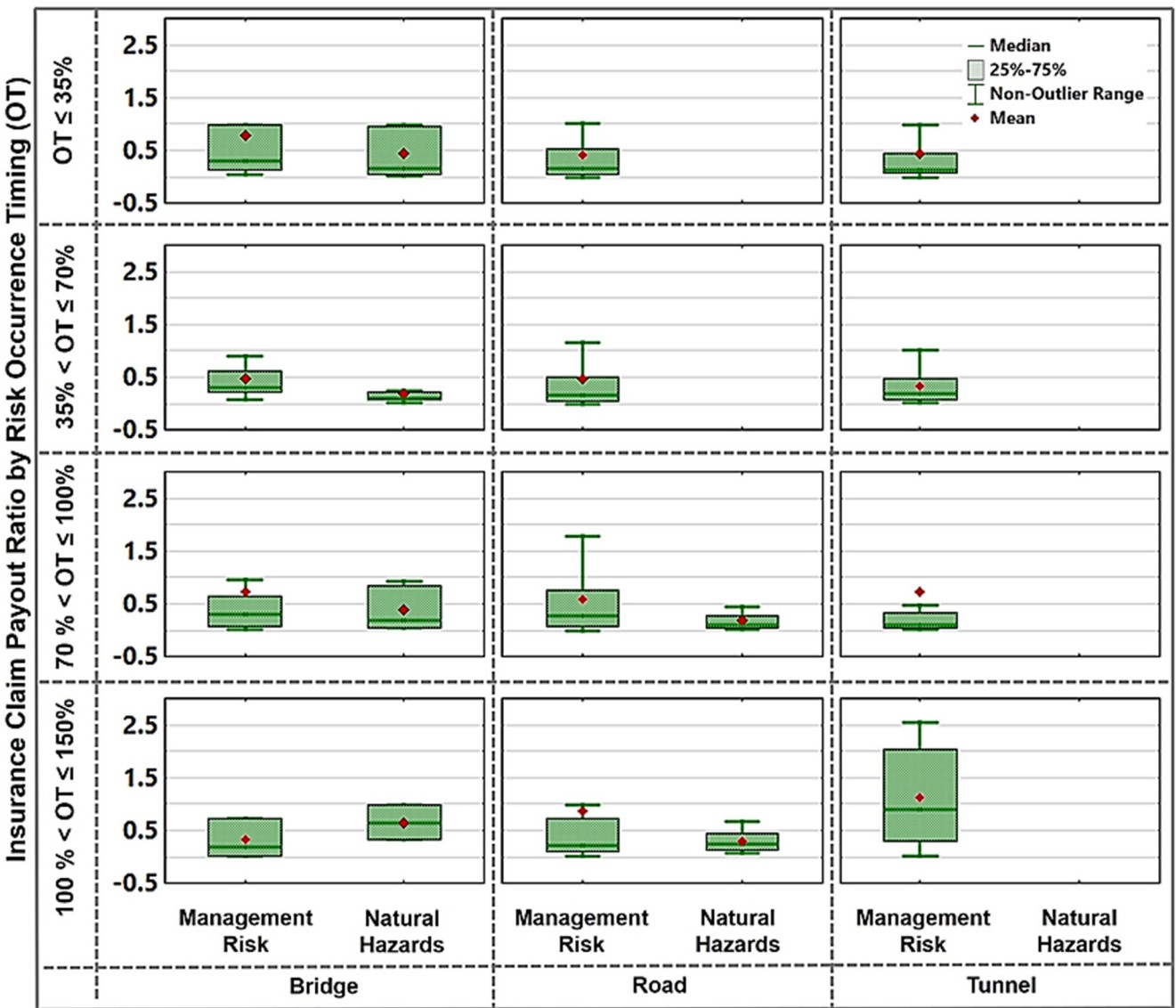

**Figure 4.** Generalized economic damage trends: claim payout ratios in the financial loss clusters by risk occurrence timing and project types.

As a way to interpret the results, detail trends in causes of financial losses could be referenced by Figure 1, in conjunction with the generalizable main outcomes of the financial loss clusters shown in Figure 4. More specifically, although bridge projects are often affected by both management risk and natural hazards compared to the other projects, the ranges of claim payout ratios were less than or similar with one maximum. This reveals that the allocated amounts of construction insurance for bridge projects are reasonably efficient during construction (OT ≤ 100%) and based on the adjusted contract time period (i.e., 100% < OT ≤ 150%).

Overall, it is recommended that the original contract end dates would be reconsidered properly for transportation infrastructure projects from the beginning of contract stage, towards sustainable project management. This is because the financial losses caused by the identified risks were investigated in all the types of projects, by showing the occurrence timing ratios between 100% and 150%. Especially, claim payout ratios associated management risk shown in tunnel projects might exceed one, which means that the allocated insurance amount might not be effective from the perspective of insurance companies.

In addition, in road construction projects, late stages of the project might be more affected by risk, especially causes included in the management risk cluster (careless, collapse,

damage, fire & explosion, slips, trips & falls, soil settlement, stolen). By comparing with the observed data shown in Figure 1, in the management risk cluster for road construction projects, the probability of damage and soil settlement would be considered as the most frequent cases causing financial losses.

## 5. Conclusions

The third-party liability insurance has been commonly adopted by the owners of large-scale construction projects, in order to cover economic damages caused by construction operation, project management, and other external risks. Among various types of construction projects, transportation infrastructure is well known as a key built asset to facilitate sustainable economic growth, but transportation construction projects are exposed to numerous unexpected events, compared to other types of construction projects. Given the nature of transportation infrastructure construction projects, many research efforts were made to predict economic damages caused by various types of risk. Although predicting financial losses associated with various risk factors is important, it is still difficult to determine which risk factors are generally critical and when these risks tend to occur, without proven references to be benchmarked. In turn, post-assessment of economic damages is crucial by classifying their generalizable patterns to develop more effective and sustainable risk management strategies for projects in the future. However, most of previous studies are prediction-focused, project type-specific, while ignoring the timing aspect of risk.

To fill these knowledge gaps, this study attempted to scientifically classify economic damages in major transportation infrastructure projects, by developing a neural network-driven machine-learning classification model that can categorize causes of financial losses depending on insurance claim payouts caused by risks and their occurrence timing in three different types of transportation infrastructure projects, such as bridges, roads, and tunnels. To this end, the insurance- and project-related data sets were collected from 625 transportation infrastructure projects completed in South Korea. The causes of financial losses were identified from historical records collected and classified into the management risk, natural hazards, and miscellaneous risk clusters. For this study, insurance claim payout proportion and risk occurrence timing were measured as economic damage indicators. Then, a total of 14 risk indicators combined with the financial loss clusters were classified into bridge, road, and tunnel projects to set the learning database. Using the BFGS back propagation training algorithm, 100 different neural network alternatives were explored incorporating 25 different combinations of activation functions for hidden and output layers into the learning process. Based on the classification accuracy in training, cross-validation, and test sets, five networks were then shortlisted. Among these alternatives, the network 9-70-3 having the highest classification accuracy was selected as the final model, which showed acceptable accuracy of 74.1%, 69.4%, and 71.8% in training, cross-validation, and test sets, respectively. Using the classification model achieved, generalizable economic damage trends were provided by claim payout ratios in each financial loss cluster associated with risk occurrence timing and project types.

This study is the first of its kind by providing benchmarkable classification references of economic damage trends in major types of transportation infrastructure projects. However, it should be noted that the limitations of this study would be considered for future work. This study was conducted based on the limited number of historical records achieved from an insurance company. With more sets of high-confidence data, the learning performance to classify the generalized patterns would be improved further. Nevertheless, the holistic approach proposed in this study will help construction practitioners consider the uncertainty of project management and the potential impact of natural hazards proactively. This study will also assist insurance companies with developing more effective and efficient financial management plans for a certain type of transportation infrastructure projects.

**Author Contributions:** Conceptualization, J.B.; methodology, J.B.; software, J.B.; validation, J.B., J.-M.K. and S.-G.Y.; formal analysis, J.B.; investigation, J.-M.K. and S.-G.Y.; resources, J.-M.K.; data curation, J.B., J.-M.K.; writing—original draft preparation, J.B.; writing—review and editing, J.B. and

J.-M.K.; visualization, J.B. and S.-G.Y.; funding acquisition, J.-M.K. All authors have read and agreed to the published version of the manuscript.

**Funding:** This work was supported by the National Research Foundation of Korea (NRF) grant funded by the Korea government (MSIT) (No. NRF-2021R1C1C2003316). ※ MSIT: Ministry of Science and ICT.

**Institutional Review Board Statement:** Not applicable.

**Informed Consent Statement:** Not applicable.

**Data Availability Statement:** The data presented in this research are available from the corresponding author by reasonable request.

**Conflicts of Interest:** The authors declare no conflict of interest.

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
