# Peer review of "Harnessing Machine Learning for Classifying Economic Damage Trends in Transportation Infrastructure Projects"

_sustainability, doi:10.3390/su13116376_

Round 1
Reviewer 1 Report
General comment
In this paper, a neural network based machine-learning classification model is developed to categorize causes of financial losses via insurance- and project-related data sets, which were obtained from completed transportation infrastructure projects in South Korea. One of the innovation is the consideration of occurrence timing (OT). The prediction results of the model are reasonable. The paper is very well written with the objectives and methods clearly described.
Detailed comment
In line 197: The network structure was firstly confirmed based on 7 input nodes and 3 output nodes, according to the variables seen in Table 1.
Please clearly list the 7 input variables here.
I recommend the paper for publication, with a decision of minor revisions. The article can be published without further review by me.
Reviewer 2 Report
The paper is well written and of interest.
I only suggest to put more into the debate on the potential use of machine learning to study economic dynamics. See the suggested references, among others.
Magazzino, C., Mele, M., & Santeramo, F. G. (2021). Using an artificial neural networks experiment to assess the links among financial development and growth in agriculture. Sustainability, 13(5), 2828.
Kim, H., Cho, H., & Ryu, D. (2020). Corporate default predictions using machine learning: Literature review. Sustainability, 12(16), 6325.
